# Psychological Wellbeing of Parents with Infants Admitted to the Neonatal Intensive Care Unit during SARS-CoV-2 Pandemic

**DOI:** 10.3390/children8090755

**Published:** 2021-08-30

**Authors:** Laura Polloni, Francesco Cavallin, Elisabetta Lolli, Rossana Schiavo, Martina Bua, Biancarosa Volpe, Marta Meneghelli, Eugenio Baraldi, Daniele Trevisanuto

**Affiliations:** 1Psychology Unit, University Hospital of Padua, 35128 Padua, Italy; laura.polloni@aopd.veneto.it (L.P.); rossana.schiavo@aopd.veneto.it (R.S.); martina.bua@aopd.veneto.it (M.B.); biancarosa.volpe@unipd.it (B.V.); 2Independent Statistician, 36020 Solagna, Italy; cescocava@libero.it; 3Neonatal Intensive Care Unit, Department of Women’s and Children’s Health, University Hospital of Padua, Via Giustiniani, 3, 35128 Padua, Italy; elisabetta.lolli@aopd.veneto.it (E.L.); meneghelli.marta@aopd.veneto.it (M.M.); eugenio.baraldi@unipd.it (E.B.)

**Keywords:** anxiety, depression, fathers, mothers, preterm child, stress

## Abstract

The current SARS-CoV-2 disease (COVID-19) pandemic is a sudden major stressor superimposed on pre-existing high distress in parents of infants admitted to the neonatal intensive care unit (NICU). This study aimed to investigate the psychological wellbeing of NICU parents during the COVID-19 pandemic. Forty-four parents of 25 inpatients of the Padua University Hospital NICU were included from June 2020 to February 2021. At 7–14 days postpartum parents completed the Edinburgh Postnatal Depression Scale (EPDS), State-Trait Anxiety Inventory (STAI), Parental Stressor Scale: NICU (PSS:NICU) and an ad-hoc questionnaire measuring parental COVID-19 related stress. About one third of parents reported extreme/high stress and a relevant negative impact on parenthood experience. Less time (82%) and less physical contact (73%) with infants due to COVID-19 preventive measures were the most frequent negative factors. Higher COVID-19 related parental stress was positively associated with anxiety, depression, NICU parental stress, stress related to NICU environment, and parental role alterations. Depression symptoms, stress related to infant condition and parental role alterations were higher in mothers. The pandemic affected parental emotional and relational wellbeing directly through additional stress due to COVID-19 concerns and indirectly through the impact of restrictions on the experience of becoming parents.

## 1. Introduction

From a psychological perspective, parents of infants admitted to the neonatal intensive care unit (NICU) are a vulnerable population, as they experience trauma (due to critical birth and separation from their infants) and stress (regarding medical conditions and related interventions) [1,2,3]. Parents report feelings of guilt, shame and high levels of stress, mood, and anxiety symptoms [1,4]. This is associated with an elevated risk for subsequent impaired child development and poorer family functioning [5,6,7,8,9]. A multi-layered approach to support parents in the NICU with particular attention to mental health has been recently highlighted [10,11]. In this context, the current corona virus disease (COVID-19) pandemic caused by the SARS-CoV-2 virus acts as a sudden, new-onset major environmental stressor superimposed on pre-existing high levels of NICU family psychological distress [3,12].

As the COVID-19 pandemic spread dangerously in Italy and worldwide, several infection preventive measures were adopted, including limitations for hospital visitors, especially in the wards with the most vulnerable patients. This forced NICU healthcare providers and parents to face severe limitations in visitation policies, challenging family-centered developmental care practices. This, in turn, may affect the ability for parents to cope with pandemic-related stress, thereby exacerbating the stress of having an infant in the NICU, and further hampering their confidence as primary caregivers [3,12,13,14].

Restrictions of visitation all together would undoubtedly add to the parental stress of NICU admission. Information on the psychological status of parents with infants admitted to a NICU during the COVID-19 pandemic is lacking. A preliminary qualitative study reported dysphoric emotions in the ten parents considered, and highlighted that restrictions accentuated the emotional and relational suffering of NICU families [13]. One of the COVID-19 preventive measures adopted at our Institution was limiting access to the NICU to one parent per baby, one hour per day. In our country, restrictions to parental visits in NICU were broadly applied but were not uniformly mandatory, thus each hospital followed local protocols and indications of the hospital directorate. Some of the NICU staff at our institution did not agree with the restrictions but could not oppose to the implementation. While strict restrictions on parental visits were initially adopted to prevent infection spread in the NICU, recent position statements indicate to relax such restrictions to favor parent-infant contact [15,16]. During the period of visit limitations, a screening of parents’ psychological wellbeing has been performed in order to identify parents in need of support. Understanding and accommodating psychological needs of NICU families seems crucial in efficiently caring for vulnerable infants in this critical historical period. This study aimed to investigate the psychological wellbeing of parents with infants admitted to the NICU during the COVID-19 pandemic. A secondary objective was to analyze the possible differences between mothers and fathers.

## 2. Materials and Methods

### 2.1. Study Design

This is an observational study on the psychological wellbeing of mothers and fathers with infants admitted to NICU after birth during the COVID-19 pandemic. The study was performed in accordance with the European regulation regarding potential sensitive data and has been approved by the Padua University Hospital Ethics Committee (protocol number 0014664). All parents gave their written consent to use the data for scientific purposes.

### 2.2. Participants

Parents of infants admitted to the NICU of the Padua Hospital during COVID-19 pandemic were eligible for inclusion. We excluded (i) parents of infants with congenital malformations or major complications, (ii) parents who suffered from a major mental illness, (iii) parents who did not understand the Italian language, and (iv) drug-user/addict mothers. Parents were included consecutively from June 2020 to February 2021. The participants were not compensated for participating in this study.

Prior to the pandemic, both parents were allowed to visit infants admitted to the NICU at the same time and without time restrictions (24/24 h). During the pandemic, access to the NICU was limited to one parent per baby, one hour per day. Such restrictions on parental visits lasted throughout the study period.

### 2.3. Data Collection

Parents were asked to complete the following questionnaires at 7–14 days postpartum. This time interval is consistent with previous studies [2,17,18,19] and EPDS instructions for administration:(1)The Edinburgh Postnatal Depression Scale (EPDS) [20,21] aims to identify depressive symptoms in women who have recently given birth. The tools have also been effectively used with fathers [2]. Individuals who score 13 or more are considered at risk of developing depression [20].(2)The State-Trait Anxiety Inventory (STAI) [22] measures symptoms of anxiety in adults. It consists of two sub-scales of 20 items respectively evaluating “state anxiety” (STAI-S anxiety in a specific situation) and “trait anxiety” (STAI-T anxiety as a general trait). Each item is based on four levels of response. A score from 40 to 50 indicates mild anxiety, 51 to 60 moderate anxiety, and >60 severe anxiety.(3)The Parental Stressor Scale: NICU (PSS: NICU) [23,24] measures stress experienced by parents during hospitalization related to parental role alteration (PRA), infant behavior and appearance (IBA), and sights and sounds of the unit (SS). Parents are asked to rate items on a five-point scale ranging from 1 (not at all stressful) to 5 (extremely stressful). An item on perceived overall parental stress is also present.(4)An ad hoc questionnaire has been developed to assess the COVID-19 related parental stress. The tool, shown in Appendix A, consists of two items asking to be answered on a Likert scale from 1 (not at all) to 5 (extremely) and one item considering the presence or absence of 10 stress factors plus an additional one that the respondents can express (*other*). The instrument focuses on the impact of the COVID-19 pandemic, in terms of stress, on the experience of becoming parent. The questions were designed based on the literature [25,26] and clinical experience. The score ranges from 2 to 21. Socio-demographic and clinical data were collected through a crosscheck between parents’ reports and medical records.

### 2.4. Statistical Analysis

Data were summarized as median and interquartile range (continuous data) or frequency and percentage (categorical data). Associations between continuous variables were assessed using Spearman’s rank correlation coefficient (r). Comparisons of continuous variables among groups were performed using the Mann-Whitney test and Kruskal-Wallis test. In the sub-analysis of mother-father pairs of the same child, continuous variables were compared in mother-father matched pairs using Wilcoxon signed rank test. All tests were two-sided and a *p*-value less than 0.05 was considered significant. Statistical analysis was performed using R 4.0 (R Foundation for Statistical Computing, Vienna, Austria) [27].

## 3. Results

During the study period, 277 newborn infants were admitted to the NICU. The parents of 113 infants were excluded because the infants had congenital malformations (n = 17) or were discharged within 8 days of life (n = 96). The parents of 117 infants were excluded because they did not understand the Italian language (n = 72) or refused to participate (n = 45). Finally, parents of 22 infants could not be contacted because the researcher was not available.

The analyses include 44 parents (19 mother-father pairs and six mothers) of 25 infants (median gestational age 34 weeks, IQR 31–35). Parental and neonatal characteristics are reported in Table 1.

COVID-19 related parental stress is displayed in Figure 1. Thirteen parents (30%) indicated the COVID-19 pandemic as an extreme/high source of stress, and 15 parents (34%) acknowledged a relevant negative impact of the COVID-19 pandemic on the experience of becoming parents. Less time (82%) and less physical contact (73%) with the infant due to preventive measures, and additional concerns about the infant’s health (57%) were the most frequent negative factors. Less support from family/friends due to preventive measures (41%), concerns about family/friends’ health (39%) and loneliness (32%) were also frequently reported. The median score on COVID-19 related parental stress was 10 out of 21 (IQR 8–12) (Appendix A).

Higher COVID-19 related parental stress was associated with a higher STAI state (r = 0.44, *p* = 0.003), higher STAI trait (r = 0.46, *p* = 0.002), higher EPDS (r = 0.35, *p* = 0.02), higher PSS-SS (r = 0.34, *p* = 0.02), higher PSS-PRA (r = 0.38, *p* = 0.01) and higher PSS-total score (r = 0.38, *p* = 0.01) (Figure 2). PSS-perceived stress item was also associated with COVID-19 related parental stress (r _s_ = 0.43, *p*=0.003). No other statistically significant associations were found between COVID-19 related parental stress and clinically relevant variables (Appendix A).

Of note, the STAI trait was higher in parents who reported a negative impact of the COVID-19 pandemic on becoming a parent due to (i) less physical contact with the infant because of the preventive measures (*p* = 0.03) and (ii) concerns about the infant’s health (*p* = 0.04) (Figure 3). The association between the main COVID-19 related factors having a negative impact on becoming a parent and the STAI trait and EPDS is reported in Appendix A.

A sub-analysis was performed in the 19 mother-father pairs (Appendix A). The median number of visits to the infant in the NICU was seven (IQR 5–12) among mothers and six (IQR 4–13) among fathers (*p* = 0.81). EPDS, PSS-IBA and PSS-PRA were higher in mothers than in fathers (*p* = 0.02, *p* = 0.01 and *p* = 0.02, respectively), while no statistically significant differences were found in COVID-19 related parental stress, STAI and PSS-SS (Figure 4). In addition, the PSS-total score was higher in mothers than in fathers (*p* = 0.01, Appendix A).

## 4. Discussion

This study aimed to investigate the psychological wellbeing of mothers and fathers with infants admitted to the NICU during the COVID-19 pandemic.

Results showed that more than half of parents scored over the cut-off for anxiety and almost one quarter scored over the cut-off for risk of depression. This is consistent with previous studies showing high levels of depression and anxiety among NICU families [2,17,19]. About one third of parents reported extreme/high stress and a relevant negative impact on the experience of becoming parents specifically due to the pandemic. Most parents reported restrictions on time and physical contact with their infant due to preventive measures as main factors that negatively impacted their experience of parenthood, alongside additional concerns for their child’s health. Limited social support from family and friends and concerns about the health of loved ones due to pandemic were other frequent responses. Interestingly, some parents spontaneously reported the inability to share visits to the child in the NICU with the partner as a further stress source.

We can observe a strong relational as well as emotional connotation in the parental stress emerging from this study. Prolonged separation from the newborn interferes with the parent/child relationship, a fundamental and neurobiologically founded factor for infant development and health [2,6]. The need of closeness, contact and interaction between parent and child is rooted in instincts programmed by evolution for the safety of the mammalian offspring, and separation may induce distress and fear in both [2,6]. During the pandemic, parental visits in NICU are limited and nurturing experiences including breastfeeding, kangaroo care, or parents’ talking with their infant, which are routinely employed to mitigate stress, are less frequent or not feasible [3,12,13]. Moreover, the inability to share NICU visits with the partner prevents parental mutual support that is an essential resource for emotional adaptation to such a distressing experience [13]. Additionally, limited social support and social isolation due to pandemic restrictions may reduce parents’ resources for coping with stress and anxiety [28,29,30].

Results showed that parental COVID-19 related stress was positively and significantly associated with STAI scores, EPDS score, PSS-total score, PSS-SS and PSS-PRA subscales, and PSS-perceived stress items. Although the cross-sectional design does not allow determining causal relationships between variables, we can observe that parents with higher the COVID-19 stress were individuals with a higher proneness to experience anxiety. People with higher anxiety traits tend to view the world as more threatening and to respond more frequently with state anxiety to situations [22,31]. During the NICU stay, the unfamiliar environment and the adverse medical conditions of the children prevent parents from immediately taking care of them and this may lead to them experiencing a higher level of stress, mostly because of parental role marginalization [17,18,23,32], as also found in the present study. Parents live in a state of physical and psychological separation from their babies feeling powerless, stressed and vulnerable to emotional difficulties [17,18,19]. The sense of powerlessness and alienation can alter parental roles and self-image, further increase negative feelings such as anxiety, depression and anger, and consequently, affect the early bond with the child [1,17,18,19,23,32,33].

This situation is undoubtedly worsened by additional pandemic worries and restrictions that limit parents/infants interactions and interfere with parental bonding and caregiving [3,12,13]. In our sample, anxiety was higher in parents who reported a negative impact of the COVID-19 pandemic on parenthood due to less physical contact with the child and additional concerns about the infant’s health due to the pandemic. Recent studies showed that concerns about COVID-19, combined with social limitations due to preventive measures, adversely affected thoughts and emotions of new mothers of full term babies, incrementing mental health symptoms such as anxiety, depression and distress [28,29,30]. Most mothers of normal newborn nursery and NICU infants reported a negative impact of the pandemic on social and family relationships, maternal role and stress, with higher degree of helplessness among mothers of NICU infants [34]. Of note, data about fathers were not available in the literature.

When comparing the psychological status of mothers and fathers, we found no differences relating to factors more linked to the context such as COVID-19 related parental stress, STAI state and PSS-SS. This suggests that parents were similarly impacted by the situation, be it the pandemic or the NICU environment. Instead, mothers and fathers showed differences in variables more related to their specific experience of parenthood such as EPDS, PSS-total score, PSS-IBA and PSS-PRA. Significantly higher scores in mothers were in line with previous studies indicating differences in parental vulnerability immediately after preterm birth [2,17,18]. In new mothers, physiological postpartum processes and hormones could affect mood and susceptibility to stress [35]. Furthermore, preterm births are often the unexpected results of mothers’ medical complications that require immediate interruption of the pregnancies to prevent serious threats to the babies and mothers’ health [17,18]. Hence, mothers’ higher EPDS score and PSS-total score could also be linked to a post-traumatic state [2,17,18,19].

Higher PSS-IBA and PSS-PRA could be explained within the framework of the “primary maternal preoccupation” [36]. It is a physiological, biologically determined special state of enhanced sensitivity that heightens parents’ abilities to anticipate the infant’s needs and signals, and constitutes a fundamental stage for the parent/infant relationship and the later infant’s cognitive, emotional and relational development [32,35,36] It develops toward the end of pregnancy and lasts for the first postnatal weeks/months. Although parents displayed a similar time course, the degree of preoccupation is significantly higher for mothers [36]. From a biological and evolutionary perspective, not being allowed or able to protect and take maternal responsibility for the infant is expected to cause emotional distress [2,32,36,37].

Despite the small sample, which represents the main limitation of this study, findings offer a first overview of the psychological wellbeing of parents of children admitted to the NICU during the pandemic. Of note, screening of the psychological wellbeing of parents was not performed before the pandemic at our institution (except for extremely few cases with mental disorders), thus comparisons with historical data were not possible.

As a consequence of the findings of this study, our institution strengthened the psychological support to parents of infants in the NICU and relaxed the restrictions on parental visits by increasing the time and the number of visits per day.

It is important to pay close attention to parents’ emotional experiences in the NICU, particularly in the context of this pandemic, since the first moments after delivery, and offer proper support especially to more vulnerable parents in order to ensure family wellbeing and prevent future problems [19]. As stress in the NICU is critical not only for parents’ mental health, but also due to potential implications for their relationship with their infant and subsequent child development, strategies are needed to mitigate it in order to optimize child health and family resilience during this incomparable period [3,12,13]. Parents’ presence in the NICU, their involvement in newborn care and the unique relationship with their baby are fundamental for infant health and neurobehavioral development as well as for parents’ wellbeing. Such aspects may be taken into consideration when limitations to visits are planned. Should restrictions be needed again in the future, adequate psychological support for parents should be provided, to respond effectively to their expected suffering [11,13,14]. Using telemedicine in an effective and equitable manner has been also recommended [12,14]. A recent study showed that video messages sent from neonatal staff to families improve parental experience and involvement, and may mitigate the effects of family separation, including during restrictions associated with COVID-19 [38].

## 5. Conclusions

The pandemic seems to affect parental wellbeing both directly and indirectly. Direct effects consist of the additional stress due to the risk of contagion and COVID-19 related concerns. Indirect effects refer to the impact of health policies and restrictions on the experience of becoming parents. The ability to deal with such a challenging situation depends on individual characteristics, resources and environmental facilitations.

## Figures and Tables

**Figure 1 children-08-00755-f001:**
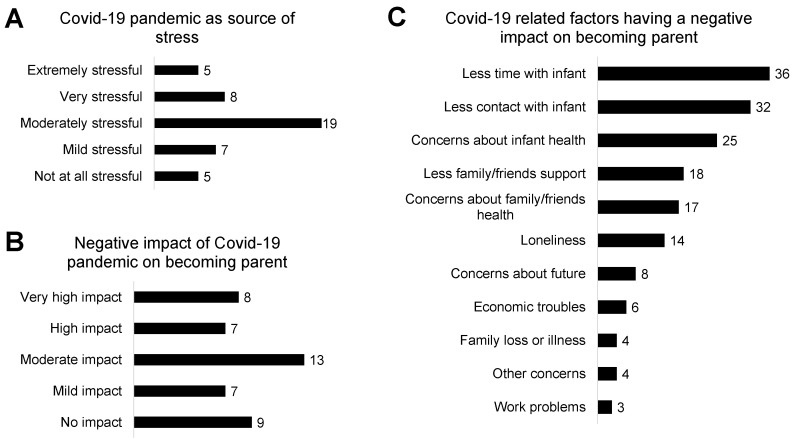
COVID-19 related parental stress: COVID-19 as source of stress (**A**), negative impact of COVID-19 on becoming a parent (**B**) and COVID-19 related factors having a negative impact on becoming a parent (**C**).

**Figure 2 children-08-00755-f002:**
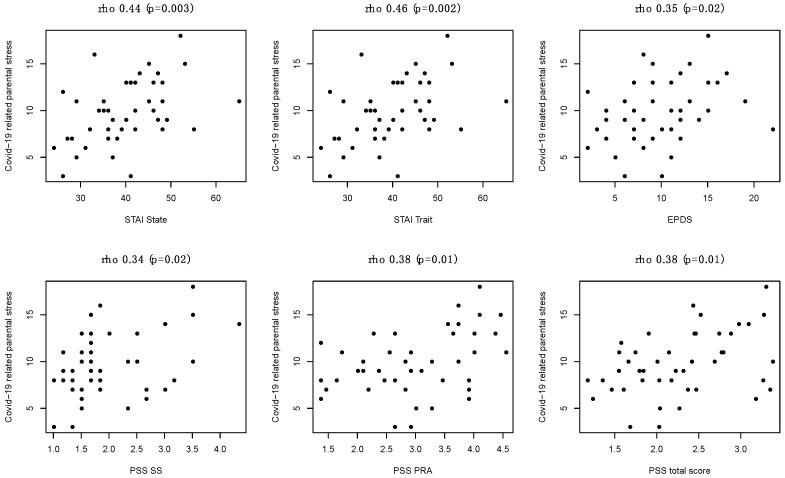
Association between COVID-19 related parental stress and STAI state, STAI trait, EPDS, PSS-SS, PSS-PRA and PSS-total score.

**Figure 3 children-08-00755-f003:**
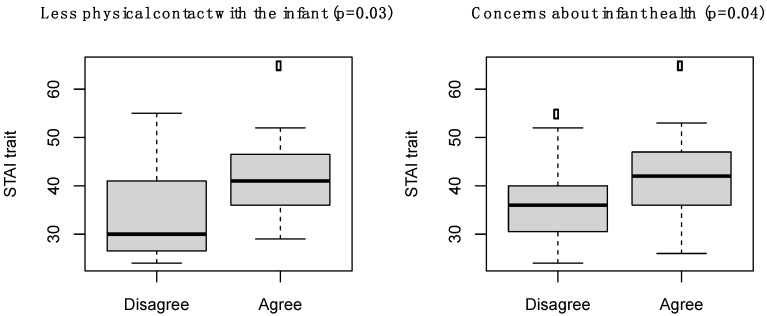
STAI trait in parents who reported a negative impact of the COVID-19 pandemic on becoming a parent due to less physical contact with the infant because of the preventive measures, and those who reported a negative impact of the COVID-19 pandemic on becoming a parent due to concerns about the infant’s health.

**Figure 4 children-08-00755-f004:**
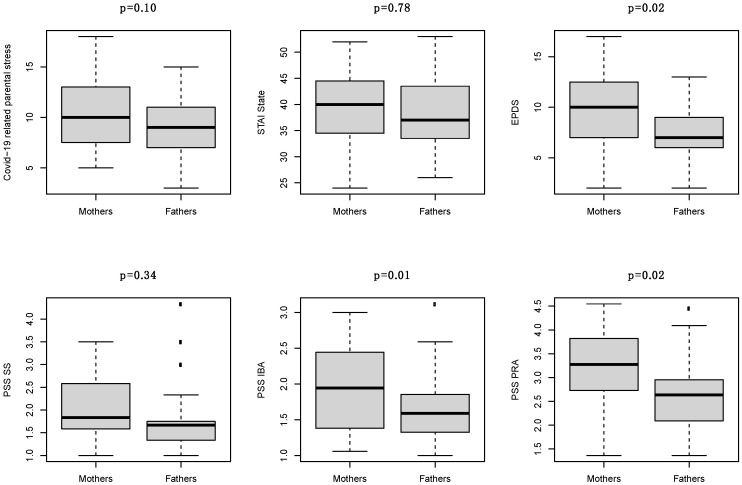
Comparison of COVID-19 related parental stress, STAI State, EPDS, PSS-SS, PSS-IBA and PSS-PRA between fathers and mothers of the same child.

**Table 1 children-08-00755-t001:** Parental and neonatal characteristics.

	Variable	N (%) or Median (IQR)
Parents (n = 44)	Parent:	
	Mothers	25 (67)
	Father	19 (43)
	Age, years	35 (32–39)
	Education level:	
	Primary or middle school	12 (27)
	High school	18 (41)
	University	14 (32)
	Number of pregnancies:	
	One pregnancy	28 (64)
	Two pregnancies	16 (36)
	Visits to the infant in the NICU, n	8 (5–14)
	STAI state	42 (37–51)
	STAI state:	
	<40	20 (46)
	40–50	12 (27)
	51–60	8 (18)
	>60	4 (9)
	STAI trait	40 (35–46)
	STAI trait:	
	<40	22 (50)
	40–50	18 (41)
	51–60	3 (7)
	>60	1 (2)
	EPDS	9 (7–12)
	EPDS ≥ 13	10 (23)
	PSS-SS	1.7 (1.5–2.4)
	PSS-IBA	1.7 (1.4–2.3)
	PSS-PRA	2.9 (2.3–3.7)
	PSS-perceived stress item	3 (3–4)
	PSS-total score	2.2 (1.8–2.7)
Infants (n = 25)	Gestational age, weeks	34 (31–35)
	Birth weight, grams	2185 (1480–3100)
	Age at parental psychological screening, days	13 (8–14)
Perinatal information (n = 25)	Conception:	
	Spontaneous	20 (80)
	Medically assisted	5 (20)
	Pregnancy:	
	Singleton	22 (88)
	Twin	3 (12)
	Delivery:	
	Vaginal or elective caesarean	6 (24)
	Emergency caesarean	19 (76)
Diagnosis at discharge (n = 25)	RDS + prematurity	16 (64)
	Sepsis	3 (12)
	Asphixia	2 (8)
	Prematurity	2 (8)
	RDS	2 (8)

EPDS: Edinburgh Postnatal Depression Scale. IBA: infant behavior and appearance. PRA: parental role alteration (PRA). PSS: parental stressor scale. SS: sights and sounds of the unit. STAI: State–Trait Anxiety Inventory.

## Data Availability

Further data are available from the corresponding author, upon reasonable request.

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
