# Peer review of "Psychological Wellbeing of Parents with Infants Admitted to the Neonatal Intensive Care Unit during SARS-CoV-2 Pandemic"

_children, 2021, doi:10.3390/children8090755_

Round 1

Reviewer 1 Report

This article is a small single-center study on the psychological impacts of COVID restrictions on NICU families. It is well written and serves to highlight the increased stress brought upon by the COVID-19 pandemic. While the description of the study and conclusions are well written, the article could be strengthened by including previously published work on the psychological impact of NICU families.

  1. While this was in a NICU, it is curious that the infants age admission in Table 1 had a median of 13 days old and ranged from 8-14 days, if I am reading it correctly. Were these infants not admitted since birth? If not, it would be worthwhile in the study design to describe the NICU at this specific hospital (what level of acuity, how many beds ,etc)

  1. In addition to describing the NICU further, a comparison between visitation policy prior to the pandemic and during the pandemic is warranted to give the readers a better understanding of the extreme changes.

  1. The average number of visits per parent seems low (IQR 4-13), but I am not sure how to interpret it necessarily since it is unknown how long the infants were admitted for. Would the authors be able to include length of stay as well?

  1. Since the study utilizes the Edinburgh PPD scale, are there historical results at the same institution that can be used as a comparison?

  1. The discussion begins with “To our knowledge, this is the first study investigating the psychological wellbeing of mothers and fathers with infants admitted to the NICU during COVID-19 pandemic.” – While this may be true at the time of conception and drafting, there seems to be at least one recent study that was similar (PMID: 33986472). The article can be strengthened by including more of these contemporary studies.

  1. The discussion should expand upon comparing the results of the study with previous results on psychological well-being of NICU families. There are some references included but perhaps this can be done in a more explicit manner.

  1. The first paragraph of the discussion can be broken up into at least 2-3 paragraphs to create more clear and concise points.

  1. The article might also be strengthened by recommendations on what one should do with the stress. How specifically can we support families better?

Author Response

We thank the Editor and the Reviewers for their comments, and we hope to have satisfactorily answered all queries.

Reviewer 1

This article is a small single-center study on the psychological impacts of COVID restrictions on NICU families. It is well written and serves to highlight the increased stress brought upon by the COVID-19 pandemic. While the description of the study and conclusions are well written, the article could be strengthened by including previously published work on the psychological impact of NICU families.

Re: We thank the Reviewer for the appreciation of our work and for the suggestions to improve the manuscript. 

  1. While this was in a NICU, it is curious that the infants age admission in Table 1 had a median of 13 days old and ranged from 8-14 days, if I am reading it correctly. Were these infants not admitted since birth? If not, it would be worthwhile in the study design to describe the NICU at this specific hospital (what level of acuity, how many beds, etc)

 Re: We thank the Reviewer for noticing it. In Table 1, the median of 13 days (IQR 8-14) actually refers to the age of the baby when parents completed the questionnaires. We corrected with "Age at parental psychological screening" in the revised manuscript (Table 1). We confirm that infants were admitted since birth: ”This is an observational study on the psychological wellbeing of mothers and fathers with infants admitted to NICU after birth during COVID-19 pandemic.” (page 2).

  1. In addition to describing the NICU further, a comparison between visitation policy prior to the pandemic and during the pandemic is warranted to give the readers a better understanding of the extreme changes.

Re: We added such information in the Methods section: “Prior to the pandemic, both parents were allowed to visit infants admitted to the NICU at the same time and without time restrictions (24/24 h). During the pandemic, access to the NICU was limited to one parent per baby, one hour per day. Such restrictions on parental visits lasted throughout the study period.” (page 2).

  1. The average number of visits per parent seems low (IQR 4-13), but I am not sure how to interpret it necessarily since it is unknown how long the infants were admitted for. Would the authors be able to include length of stay as well?

Re: The access to the NICU was limited to one parent per baby, one hour per day. The median number of visits to the infant in the NICU for each parent was 8 (IQR 5-14) with a median duration of hospitalization at the time of parental screening of 13 days (IQR 8-14).

  1. Since the study utilizes the Edinburgh PPD scale, are there historical results at the same institution that can be used as a comparison?

Re: Unfortunately, there was no screening of the psychological wellbeing of parents before the pandemic, except for very few cases with mental disorders. Therefore, we cannot perform any comparisons with historical results at the same institution. We added this consideration in the Discussion section: “Of note, screening of the psychological wellbeing of parents was not performed before the pandemic at our institution (except for very few cases with mental disorders, thus comparisons with historical data were not possible.” (page 10). We hope that the present study will raise awareness among institutions on the importance of paying attention to the psychological wellbeing of parents in their normal clinical routine.

  1. The discussion begins with “To our knowledge, this is the first study investigating the psychological wellbeing of mothers and fathers with infants admitted to the NICU during COVID-19 pandemic.” – While this may be true at the time of conception and drafting, there seems to be at least one recent study that was similar (PMID: 33986472). The article can be strengthened by including more of these contemporary studies.

Re: We agree with the Reviewer about the rapidly growing topic. As suggested, we run a new literature search and found the original article suggested by the Reviewer (new ref #34) and one review (new ref #14). We included these contemporary studies in the revised manuscript. We modified the first sentence of the Discussion as “This study aimed to investigate the psychological wellbeing of mothers and fathers with infants admitted to the NICU during COVID-19 pandemic.” (page 7). We added a comment in the Discussion: “Most mothers of normal newborn nursery and NICU infants reported a negative impact of the pandemic on social and family relationships, maternal role and stress, with higher degree of helplessness among mothers of NICU infants. [34] Of note, data about fathers were not available in the literature.” (page 9).

  1. The discussion should expand upon comparing the results of the study with previous results on psychological well-being of NICU families. There are some references included but perhaps this can be done in a more explicit manner.

Re: We thank the Reviewer for the suggestion. In addition to the considerations already included in the Discussion section at page 9 (“Significantly higher scores in mothers were in line with previous studies indicating differences in parental vulnerability immediately after preterm birth [2], [15], [16]” and “During the NICU stay, the unfamiliar environment and the adverse medical conditions of the children prevent parents from immediately taking care of them and this may lead to experience higher level of stress, mostly because of parental role marginalization [15], [16], [21], [30], as also found in the present study.”), we added a further comment: “Results showed that more than half of parents scored over the cut-off for anxiety and almost one quarter scored over the cut-off for risk of depression. This is consistent with previous studies showing high levels of depression and anxiety among NICU families [2], [15], [17].” (Discussion, page 8).

  1. The first paragraph of the discussion can be broken up into at least 2-3 paragraphs to create more clear and concise points.

Re: We thank the Reviewer for the suggestion, which improved the readability of the Discussion section. In the revised manuscript, we broke up the first paragraph of the discussion into more paragraphs.

  1. The article might also be strengthened by recommendations on what one should do with the stress. How specifically can we support families better?

Re: The original version included the recommendation that “adequate psychological support for parents should be provided, to respond effectively to their expected suffering [11], [13]” (Discussion, page 10). In the revised version, we added this further comment: “Using telemedicine in an effective and equitable manner has been also recommended [12], [14]. A recent study showed that video messages sent from neonatal staff to families improve parental experience and involvement, and may mitigate the effects of family separation, including during restrictions associated with COVID-19 [36]” (Discussion, page 10).

Reviewer 2 Report

The authors report on a small study on 44 parents (related to 25 neonates) with a single stress assessment at 7-14 days. The findings ‘confirm’ the anticipated high (dis)stress and the fact that less time and less physical contact were reported as specific burdens.

I understand that restrictions on parental visits were imposed, but to the very best of my knowledge, this practice is only very poorly supported by evidence, with claims to lift these practices have been reported. I therefore miss somewhat the balance in the introduction. This should be further explored and discussed (eg EFCNI and covid).

What’s the rationale for the 7-14 days time interval.

I assume that the 25 cases are only a minority of cases admitted. At least, the authors should provide information on the total population, consent/not consented/not asked. We also need some more information on the disease severity of the included cases (it seems that the medical issues were rather limited, eg gestational age).

How does these findings compare to non covid setting ?

This is a fast growing field of research, so that I highly recommend to check PubMed again on ‘similar observations, like https://pubmed.ncbi.nlm.nih.gov/?term=COVID+nicu+parent+stress&sort=date, this link does provide relevant reports for your paper.

Editing comment: The references should be checked on their format.

Author Response

We thank the Editor and the Reviewers for their comments, and we hope to have satisfactorily answered to all queries.

Reviewer 2

  1. The authors report on a small study on 44 parents (related to 25 neonates) with a single stress assessment at 7-14 days. The findings ‘confirm’ the anticipated high (dis)stress and the fact that less time and less physical contact were reported as specific burdens. I understand that restrictions on parental visits were imposed, but to the very best of my knowledge, this practice is only very poorly supported by evidence, with claims to lift these practices have been reported. I therefore miss somewhat the balance in the introduction. This should be further explored and discussed (eg EFCNI and covid).

Re: We thank the Reviewer for the comment. We added a consideration in the Introduction “While strict restrictions on parental visits were initially adopted to prevent infection spread in the NICU, recent position statements indicate to relax such restrictions to favor parent-infant contact.” (page 2) with reference to the indications by SIN (Italian Society of Neonatology) and EFCNI (ref #15  and #16).

  1. What’s the rationale for the 7-14 days time interval.

Re: The choice of this time interval was based on previous studies on parental psychological wellbeing in the NICU setting (ref #2, #27, #28 and #30), and on the fact that the EPDS inquires the intensity of depressive symptoms over the preceding seven days. We specified this aspect in the Data collection subsection as follows: “This time interval is consistent with previous studies [2], [15]–[17] and EPDS instruction for administration” (Methods, page 2).

  1. I assume that the 25 cases are only a minority of cases admitted. At least, the authors should provide information on the total population, consent/not consented/not asked. We also need some more information on the disease severity of the included cases (it seems that the medical issues were rather limited, eg gestational age).

Re: We added some information about participant inclusion: “During the study period, 277 newborn infants were admitted to the NICU. The parents of 113 infants were excluded because the infants had congenital malformations (n=17) or were discharged within 8 days of life (n=96). The parents of 117 infants were excluded because they did not understand Italian language (n=72) or refused to participate (n=45). Finally, parents of 22 infants could not be contacted because the researcher was not available. The analyses include 44 parents (19 mother-father pairs and six mothers) of 25 infants (median gestational age 34 weeks, IQR 31-35). Parental and neonatal characteristics are reported in Table 1.” (Results, pages 4-5).

We added the diagnosis at discharge in Table 1.

  1. How does these findings compare to non-covid setting?

Re: In addition to the considerations already included in the Discussion section at page 9 (“Significantly higher scores in mothers were in line with previous studies indicating differences in parental vulnerability immediately after preterm birth [2], [15], [16]” and “During the NICU stay, the unfamiliar environment and the adverse medical conditions of the children prevent parents from immediately taking care of them and this may lead to experience higher level of stress, mostly because of parental role marginalization [15], [16], [21], [30], as also found in the present study.”), we added a further comment: “Results showed that more than half of parents scored over the cut-off for anxiety and almost one quarter scored over the cut-off for risk of depression. This is consistent with previous studies showing high levels of depression and anxiety among NICU families [2], [15], [17].” (Discussion, page 8).

  1. This is a fast growing field of research, so that I highly recommend to check PubMed again on ‘similar observations, like https://pubmed.ncbi.nlm.nih.gov/?term=COVID+nicu+parent+stress&sort=date, this link does provide relevant reports for your paper.

Re: We agree with the Reviewer about the rapidly growing topic. As suggested, we run a new literature search and found two additional references, which were pertinent to the topic and added in the revised manuscript (new ref #14, #34).

The article by Erdei et al. and the article by Bembich were already included in the Introduction section (ref #3 and #13).

In the Discussion, we removed the statement about our paper being the first study investigating psychological wellbeing of mothers and fathers with infants admitted to the NICU during COVID-19 pandemic, and we modified the first sentence of the Discussion as “This study aimed to investigate the psychological wellbeing of mothers and fathers with infants admitted to the NICU during COVID-19 pandemic.” (page 7).

We added a comment in the Discussion: “Most mothers of normal newborn nursery and NICU infants reported a negative impact of the pandemic on social and family relationships, maternal role and stress, with higher degree of helplessness among mothers of NICU infants. [34] Of note, data about fathers were not available in the literature.” (page 9).

  1. Editing comment: The references should be checked on their format.

Re: We thank the Reviewer for the suggestion. We check the references and corrected the format of ref #32 #35 and #36 on page 9.

Reviewer 3 Report

Overview: This is a well written research article on psychological wellbeing of parents which infants are admitted to the NICU during the COVID-19 pandemic. The topic is of interest and is very pertinent.

Major comment:

1) Based on the results of this study, did the author's institution change their practices? What was the basis for such intense and strict restrictions in the NICU at the author's institution?

2) Although there is a discussion of the problem, the alternative approaches are not discussed adequately. Can telemedicine solve this issue? 

Abstract:

Line 14: “acts as” can be replaced by “is” to simplify the sentence.

Line 16: “the” may be replaced by “this”.

Introduction:

Lines 32-35: sentence is very long, consider splitting into 2 sentences.

Line 35: “and” between guilt and shame can be removed.

Line 38: Consider rephrasing the sentence as “A multi-layered approach to support parents in the NICU with particular attention to mental health has been recently highlighted”

Line 40: “SARS-CoV-2 disease (COVID-19) pandemic” can be rephrased as “ Corona virus disease 2019 (COVID-19) pandemic caused by the SARS-CoV-2 virus”.  

Line 44: “control” may be removed.

Lines 55-57: how do the restrictions in the author’s institution compare to the restrictions in the rest of the country? If different, why were the restrictions different? Did the staff agree on the restrictions or was there any opposition? Did any difference lead to change in restrictions?

Line 58: Consider removing “early” or rephrasing the sentence.

Line 60: “crucial to efficiently take care of vulnerable infants”- consider replacing with “crucial in efficiently caring for vulnerable infants”

Line 70: were the participants compensated in any way for participating in this study?

Line 72: NICU “in or at” Padua Hospital- in or at is missing.

Lines 74-76: Since parents are included or excluded, why do lines 74-75 talk about including excluding “infants”?

Line 77: Were the restrictions kept the same during the study period or were they modified during the study period?

Line 79: Why was 7-14 days post-partum chosen as the timing for the questionnaires?

Table 1: A footnote clarifying and expanding the abbreviations is missing.

Line 198: “artificial environment”- what do the authors mean by artificial environment.

Line 205: “It” is nor clear in this sentence. “happens” should be happen.

Author Response

We thank the Editor and the Reviewers for their comments, and we hope to have satisfactorily answered all queries.

Reviewer 3

Overview: This is a well written research article on psychological wellbeing of parents which infants are admitted to the NICU during the COVID-19 pandemic. The topic is of interest and is very pertinent.

Re: We thank the Reviewer for the appreciation of our work and for the suggestions to improve the manuscript. 

  1. Based on the results of this study, did the author's institution change their practices? What was the basis for such intense and strict restrictions in the NICU at the author's institution?

Re: The initial implementation of the strict restrictions in the NICU at our institution aimed at reducing the risk of contagion to a very vulnerable population (infants in the NICU). We are aware that recent claims to lift these practices have been reported and we added a consideration in the Introduction “While strict restrictions on parental visits were initially adopted to prevent infection spread in the NICU, recent position statements indicate to relax such restrictions to favor parent-infant contact.” (page 2). We also included some considerations about institutional changes to practices due to our findings: “As a consequence of the findings of this study, our institution strengthened the psychological support to parents of infants in NICU and relaxed the restrictions on parental visits by increasing the time and the number of visits per day.” (page 10).

  1. Although there is a discussion of the problem, the alternative approaches are not discussed adequately. Can telemedicine solve this issue?

Re: In addition to the recommendation that “adequate psychological support for parents should be provided, to respond effectively to their expected suffering [11], [13] [14]” reported at the end of the discussion, we added this further comment: “Using telemedicine in an effective and equitable manner has been also recommended [12], [14]. A recent study showed that video messages sent from neonatal staff to families improve parental experience and involvement, and may mitigate the effects of family separation, including during restrictions associated with COVID-19 [38]” (Discussion, page 10).

  1. Abstract: Line 14: “acts as” can be replaced by “is” to simplify the sentence.

Re: We replaced “acts as” with “is” (Abstract, page 1).

  1. Abstract: Line 16: “the” may be replaced by “this”.

Re: We replaced “the” with “this” (Abstract, page 1).

  1. Lines 32-35: sentence is very long, consider splitting into 2 sentences.

Re: We prefer to keep it as single sentence, but we acknowledge that the sentence in the original version was difficult to read. In the revised version, we rephrased it as “From a psychological perspective, parents of infants admitted to the neonatal intensive care unit (NICU) are a vulnerable population, as they experience trauma (due to critical birth and separation from their infants) and stress (regarding medical conditions and related interventions) [1]–[3].” (page 1) to improve readability for the reader.

  1. Line 35: “and” between guilt and shame can be removed.

Re: We removed “and” in line 35 of the Introduction (page 1).

  1. Line 38: Consider rephrasing the sentence as “A multi-layered approach to support parents in the NICU with particular attention to mental health has been recently highlighted”

Re: In the revised version, we rephrased the sentence as suggested (page 1).

  1. Line 40: “SARS-CoV-2 disease (COVID-19) pandemic” can be rephrased as “Corona virus disease 2019 (COVID-19) pandemic caused by the SARS-CoV-2 virus”.

Re: In the revised version, we rephrased the sentence as suggested (page 1).

  1. Line 44: “control” may be removed.

Re: We removed “control”.

  1. Lines 55-57: how do the restrictions in the author’s institution compare to the restrictions in the rest of the country? If different, why were the restrictions different? Did the staff agree on the restrictions or was there any opposition? Did any difference lead to change in restrictions?

Re: We added these considerations in the Introduction: “In our country, restrictions to parental visits in NICU were broadly applied but were not uniformly mandatory, thus each hospital followed local protocols and indications of the hospital directorate. Some of the NICU staff at our institution did not agree with the restrictions but could not oppose to the implementation.” (page 2).

  1. Line 58: Consider removing “early” or rephrasing the sentence.

Re: We removed “early” from that sentence (page 2).

  1. Line 60: “crucial to efficiently take care of vulnerable infants”- consider replacing with “crucial in efficiently caring for vulnerable infants”

Re: We change the sentence as suggested (page 2).

  1. Line 70: were the participants compensated in any way for participating in this study?

Re: We added “The participants were not compensated for participating in this study.” In the revised manuscript (page 2).

  1. Line 72: NICU “in or at” Padua Hospital- in or at is missing.

Re: We revise the sentence as “Parents of infants admitted to the NICU of the Padua Hospital during COVID-19 pandemic were eligible for inclusion.” (page 2).

  1. Lines 74-76: Since parents are included or excluded, why do lines 74-75 talk about including excluding “infants”?

Re: We rephrased the sentence about exclusion criteria as “We excluded i) parents of infants with congenital malformations or major complications, ii) parents who suffered from a major mental illness, iii) parents who did not understand Italian language, and iv) drug-user/addict mothers.” (page 2).

  1. Line 77: Were the restrictions kept the same during the study period or were they modified during the study period?

Re: The restrictions were in place during the study period and were not modified. In the Methods section, we specified that “During the pandemic, access to the NICU was limited to one parent per baby, one hour per day. Such restrictions on parental visits lasted throughout the study period.” (Participants subsection, page 2).

  1. Line 79: Why was 7-14 days post-partum chosen as the timing for the questionnaires?

Re: We chose this time interval based on previous studies on parental psychological wellbeing in the NICU setting (e.g. ref #2, #17-19), and on the fact that the EPDS asks the intensity of depressive symptoms over the preceding seven days. We specified it in the Data collection subsection as follows: “This time interval is consistent with previous studies [2], [17]–[19] and EPDS instructions for administration” (page 2).

  1. Table 1: A footnote clarifying and expanding the abbreviations is missing.

Re: We added the footnote as indicated.

  1. Line 198: “artificial environment”- what do the authors mean by artificial environment.

Re: We removed the term "artificial" and kept the word "unfamiliar" (page 9).

  1. Line 205: “It” is nor clear in this sentence. “happens” should be happen.

Re: We removed that sentence from the revised manuscript.

Round 2

Reviewer 2 Report

no additional comments